# Matching neural paths: transfer from recognition to correspondence search

Nikolay Savinov[1]          Lubor Ladicky[1]          Marc Pollefeys[1,2]

[1]Department of Computer Science at ETH Zurich, [2]Microsoft
{nikolay.savinov,lubor.ladicky,marc.pollefeys}@inf.ethz.ch

## Abstract

Many machine learning tasks require finding per-part correspondences between objects. In this work we focus on low-level correspondences — a highly ambiguous matching problem. We propose to use a hierarchical semantic representation of the objects, coming from a convolutional neural network, to solve this ambiguity. Training it for low-level correspondence prediction directly might not be an option in some domains where the ground-truth correspondences are hard to obtain. We show how transfer from recognition can be used to avoid such training. Our idea is to mark parts as "matching" if their features are close to each other at all the levels of convolutional feature hierarchy (neural paths). Although the overall number of such paths is exponential in the number of layers, we propose a polynomial algorithm for aggregating all of them in a single backward pass. The empirical validation is done on the task of stereo correspondence and demonstrates that we achieve competitive results among the methods which do not use labeled target domain data.

## 1   Introduction

Finding per-part correspondences between objects is a long-standing problem in machine learning. The level at which correspondences are established can go as low as pixels for images or millisecond timestamps for sound signals. Typically, it is highly ambiguous to match at such a low level: a pixel or a timestamp just does not contain enough information to be discriminative and many false positives will follow. A hierarchical semantic representation could help to solve the ambiguity: we could choose the low-level match which also matches at the higher levels. For example, a car contains a wheel which contains a bolt. If we want to check if this bolt matches the bolt in another view of the car, we should check if the wheel and the car match as well.

One possible hierarchical semantic representation could be computed by a convolutional neural network. The features in such a network are composed in a hierarchical manner: the lower-level features are used to compute higher-level features by applying convolutions, max-poolings and non-linear activation functions on them. Nevertheless, training such a convolutional neural network for correspondence prediction directly (e.g., [25], [2]) might not be an option in some domains where the ground-truth correspondences are hard and expensive to obtain. This raises the question of scalability of such approaches and motivates the search for methods which do not require training correspondence data.

To address the training data problem, we could transfer the knowledge from the source domain where the labels are present to the target domain where no labels or few labeled data are present. The most common form of transfer is from classification tasks. Its promise is two-fold. First, classification labels are one of the easiest to obtain as it is a natural task for humans. This allows to create huge recognition datasets like Imagenet [18]. Second, the features from the low to mid-levels have been shown to transfer well to a variety of tasks [22], [3], [15].

Although there has been a huge progress in transfer from classification to detection [7], [17], [19], [16], segmentation [12], [1] and other semantic reasoning tasks like single-image depth prediction [4], the transfer to correspondence search has been limited [13], [10], [8].

We propose a general solution to unsupervised transfer from recognition to correspondence search at the lowest level (pixels, sound millisecond timestamps). Our approach is to match paths of activations coming from a convolutional neural network, applied on two objects to be matched. More precisely, to establish matching on the lowest level, we require the features to match at all different levels of convolutional feature hierarchy. Those different-level features form paths. One such path would consist of neural activations reachable from the lowest-level feature to the highest-level feature in the network topology (in other words, the lowest level feature lies in the receptive field of the highest level). Since every lowest-level feature belongs to many paths, we do voting based on all of them.

Although the overall number of such paths is exponential in the number of layers and thus infeasible to compute naively, we prove that the voting is possible in polynomial time in a single backward pass through the network. The algorithm is based on dynamic programming and is similar to the backward pass for gradient computation in the neural network.

Empirical validation is done on the task of stereo correspondence on two datasets: KITTI 2012 [6] and KITTI 2015 [14]. We quantitatively show that our method is competitive among the methods which do not require labeled target domain data. We also qualitatively show that even dramatic changes in low-level structure can be handled reasonably by our method due to the robustness of the recognition hierarchy: we apply different style transfers [5] to corresponding images in KITTI 2015 and still successfully find correspondences.

## 2 Notation

Our method is generally applicable to the cases where the input data has a multi-dimensional grid topology layout. We will assume input objects $o$ to be from the set of $B$-dimensional grids $\Phi \subset \mathbb{R}^B$ and run convolutional neural networks on those grids. The per-layer activations from those networks will be contained in the set of $(B+1)$-dimensional grids $\Psi \subset \mathbb{R}^{B+1}$. Both the input data and the activations will be indexed by a $(B+1)$-dimensional vector $\mathbf{x} = (x, y, \ldots, c) \in \mathbb{N}^{B+1}$, where $x$ is a column index, $y$ is a row index, etc., and $c \in \{1, \ldots, C\}$ is the channel index (we will assume $C = 1$ for the input data, which is a non-restrictive assumption as we will explain later).

We will search for correspondences between those grids, thus our goal will be to estimate shifts $\mathbf{d} \in \mathcal{D} \subset \mathbb{Z}^{B+1}$ for all elements in the grid. The choice of the shift set $\mathcal{D}$ is task-dependent. For example, for sound $B = 1$ and only 1D shifts can be considered. For images, $B = 2$ and $\mathcal{D}$ could be a set of 1D shifts (usually called a stereo task) or a set of 2D shifts (usually called an optical flow task).

In this work, we will be dealing with convolutional neural network architectures, consisting of convolutions, max-poolings and non-linear activation functions (one example of such an architecture is a VGG-net [20], if we omit softmax which we will not use for the transfer). We assume every convolutional layer to be followed by a non-linear activation function throughout the paper and will not specify those functions explicitly.

The computational graph of these architectures is a directed acyclic graph $G = \{A, E\}$, where $A = \{a_1, \ldots, a_{|A|}\}$ is a set of nodes, corresponding to neuron activations ($|A|$ denotes the size of this set), and $E = \{e_1, \ldots, e_{|E|}\}$ is a set of arcs, corresponding to computational dependencies ($|E|$ denotes the size of this set). Each arc is represented as a tuple $(a_i, a_j)$, where $a_i$ is the input (origin), $a_j$ is the output (endpoint). The node set consists of disjoint layers $A = \bigcup_{\ell=0}^{L} A_\ell$. The arcs are only allowed to go from the previous layer to the next one.

We will use the notation $A_\ell(\mathbf{x})$ for the node in $\ell$-th layer at position $\mathbf{x}$; $\mathbf{in}(\mathbf{x}_\ell)$ for the set of origins $\mathbf{x}_{\ell-1}$ of arcs, entering layer $\ell$ at position $\mathbf{x}_\ell$ of the reference object; $\mathbf{x}_{\ell+1} \in \mathbf{out}(\mathbf{x}_\ell)$ for the set of endpoints of arcs, exiting layer $\ell$ at position $\mathbf{x}_\ell$ of the reference object. Let $f_\ell \in F = \{\texttt{maxpool}, \texttt{conv}\}$ be the mathematic operator which corresponds to forward computation in layer $\ell$ as $a \leftarrow f_\ell(\mathbf{in}(a))$, $a \in A_\ell$ (with a slight abuse of notation, we use $a$ for both the nodes in the computational graph and the activation values which are computed in those nodes).

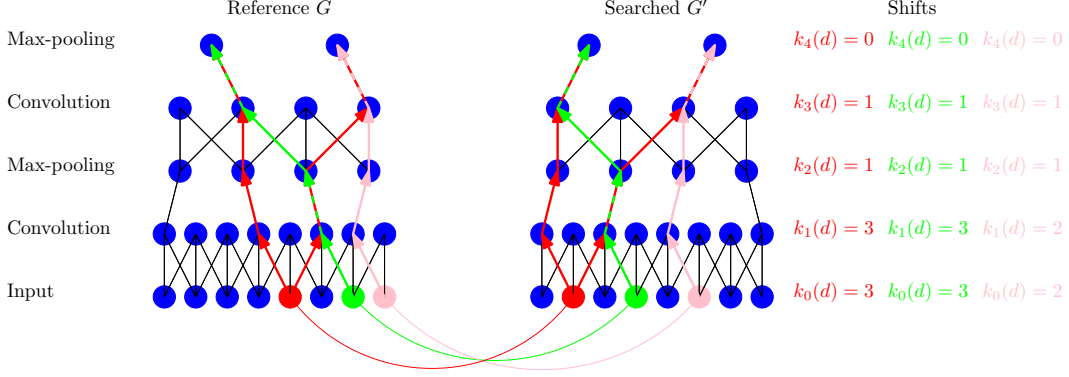

Figure 1: Four siamese paths are shown. Two of them (red) have the same origin and support the hypothesis of the shift $d = 3$ for this origin. The other two (green and pink) have different origins and support hypotheses $d = 3$ and $d = 2$ for their respective origins.

## 3    Correspondence via path matching

We will consider two objects, reference $o \in \Phi$ and searched $o' \in \Phi$, for which we want to find correspondences. After applying a CNN on them, we get graphs $G$ and $G'$ of activations. The goal is to establish correspondences between the input-data layers $A_0$ and $A_0'$. That is, every cell $A_0(\mathbf{x})$ in the reference object $o \in \Phi$ has a certain shift $\mathbf{d} \in \mathcal{D}$ in the searched object $o' \in \Phi$, and we want to estimate $\mathbf{d}$.

Here comes the cornerstone idea of our method: we establish the matching of $A_0(\mathbf{x})$ with $A_0'(\mathbf{x} - \mathbf{d})$ for a shift $\mathbf{d}$ if there is a pair of "parallel" paths (we call this pair a siamese path), originating at those nodes and ending at the last layers $A_L$, $A_L'$, which match. This pair of paths must have the same spatial shift with respect to each other at all layers, up to subsampling, and go through the same feature channels with respect to each other. We take the subsampling into account by per-layer functions

$$k_\ell(\mathbf{d}) = \gamma_\ell(k_{\ell-1}(\mathbf{d})), \ell = 1, \dots, L, \qquad \gamma_\ell(\tilde{\mathbf{d}}) = \left\lfloor \frac{\tilde{\mathbf{d}}}{q_\ell} \right\rfloor, \qquad k_0(\mathbf{d}) = \mathbf{d}, \qquad (1)$$

where $k_\ell(\mathbf{d})$ is how the zero-layer shift $\mathbf{d}$ transforms at layer $\ell$, $q_\ell$ is the $\ell$-th layer spatial subsampling factor (note that rounding and division on vectors is done element-wise). Then a siamese path $P$ can be represented as

$$P = (p, p'), \quad p = (A_0(\mathbf{x}_0^P), \dots, A_L(\mathbf{x}_L^P)), \quad p' = (A_0'(\mathbf{x}_0^P - k_0(\mathbf{d})), \dots, A_L'(\mathbf{x}_L^P - k_L(\mathbf{d}))) \qquad (2)$$

where $\mathbf{x}_0^P = \mathbf{x}$ and $\mathbf{x}_\ell^P$ denotes the position at which the path $P$ intersects layer $\ell$ of the reference activation graph. Such paths are illustrated in Fig. 1. The logic is simple: matching in a siamese path means that the recognition hierarchy detects the same features at different perception levels with the same shifts (up to subsampling) with respect to the currently estimated position $\mathbf{x}$, which allows for a confident prediction of match. The fact that a siamese path is "matched" can be established by computing the matching function (high if it matches, low if not)

$$M(P) = \bigodot_{\ell=0}^{L} m_\ell(A_\ell(\mathbf{x}_\ell^P), A_\ell'(\mathbf{x}_\ell^P - k_\ell(\mathbf{d}))) \qquad (3)$$

where $m_\ell(\cdot, \cdot)$ is a matching function for individual neurons (prefers them both to be similar and non-zero at the same time) and $\odot$ is a logical-and-like operator. Both will be discussed later.

Since we want to estimate the shift for a node $A_0(\mathbf{x})$, we will consider all possible shifts and vote for each of them. Let us denote a set of siamese paths, starting at $A_\ell(\mathbf{x})$ and $A_\ell'(\mathbf{x} - \mathbf{d})$ and ending at the last layer, as $\mathcal{P}_\ell(\mathbf{x}, \mathbf{d})$.

For every shift $\mathbf{d} \in \mathcal{D}$ we introduce $U(\mathbf{x}, \mathbf{d})$ as the log-likelihood of the event that $\mathbf{d}$ is the correct shift, i.e. $A_0(\mathbf{x})$ matches $A_0'(\mathbf{x} - \mathbf{d})$. To collect the evidence from all possible paths, we "sum up"

the matching functions for all individual paths, leading to

$$U(\mathbf{x}, \mathbf{d}) = \bigoplus_{P \in \mathcal{P}_0(\mathbf{x}, \mathbf{d})} M(P) = \bigoplus_{P \in \mathcal{P}_0(\mathbf{x}, \mathbf{d})} \bigodot_{\ell=0}^{L} m_\ell(A_\ell(\mathbf{x}_\ell^P), A'_\ell(\mathbf{x}_\ell^P - k_\ell(\mathbf{d}))) \qquad (4)$$

where the sum-like operator $\oplus$ will be discussed later.

The distribution $U(\mathbf{x}, \mathbf{d})$ can be used to either obtain the solution as $\mathbf{d}^*(\mathbf{x}) = \arg\max_{\mathbf{d} \in \mathcal{D}} U(\mathbf{x}, \mathbf{d})$ or to post-process the distribution with any kind of spatial smoothing optimization and then again take the best-cost solution.

The obvious obstacle to using the distribution $U(\mathbf{x}, \mathbf{d})$ is that

**Observation 1.** *If $K$ is the minimal number of activation channels in all the layers of the network and $L$ is the number of layers, the number of paths, considered in the computation of $U(\mathbf{x}, \mathbf{d})$ for a single originating node, is $\Omega(K^L)$ — at least exponential in the number of layers.*

In practice, it is infeasible to compute $U(\mathbf{x}, \mathbf{d})$ naively. In this work, we prove that it is possible to compute $U(\mathbf{x}, \mathbf{d})$ in $O(|A| + |E|)$ — thus linear in the number of layers — using the algorithm which will be introduced in the next section.

## 4   Linear-time backward algorithm

**Theorem 1.** *For any $m_\ell(\cdot, \cdot)$ and any pair of operators $\langle \oplus, \odot \rangle$ such that $\odot$ is left-distributive over $\oplus$, i.e. $a \odot (b \oplus c) = a \odot b \oplus a \odot c$, we can compute $U(\mathbf{x}, \mathbf{d})$ for all $\mathbf{x}$ and $\mathbf{d}$ in $O(|A| + |E|)$.*

**Proof** Since there is distributivity, we can use a dynamic programming approach similar to the one developed for gradient backpropagation.

First, let us introduce subsampling functions $k_s^\ell(\mathbf{d}) = \gamma_s(k_{s-1}^\ell(\mathbf{d})), k_\ell^\ell(\mathbf{d}) = \mathbf{d}, s \geq \ell$. Note that $k_s^0 = k_s$ as introduced in Eq. 1.

Then, let us introduce auxiliary variables $U_\ell(\mathbf{x}_\ell, \mathbf{d})$ for each layer $\ell = 0, \ldots, L$, which have the same definition as $U(\mathbf{x}, \mathbf{d})$ except for the fact that the paths, considered in them, start from the later layer $\ell$:

$$U_\ell(\mathbf{x}_\ell, \mathbf{d}) = \bigoplus_{P \in \mathcal{P}_\ell(\mathbf{x}_\ell, \mathbf{d})} M(P) = \bigoplus_{P \in \mathcal{P}_\ell(\mathbf{x}_\ell, \mathbf{d})} \bigodot_{s=\ell}^{L} m_s(A_s(\mathbf{x}_s^P), A'_s(\mathbf{x}_s^P - k_s^\ell(\mathbf{d}))). \qquad (5)$$

Note that $U(\mathbf{x}, \mathbf{d}) = U_0(\mathbf{x}, \mathbf{d})$. The idea is to iteratively recompute $U_\ell(\mathbf{x}_\ell, \mathbf{d})$ based on known $U_{\ell+1}(\mathbf{x}_{\ell+1}, \gamma_\ell(\mathbf{d}))$ for all $\mathbf{x}_{\ell+1}$. Eventually, we will get to the desired $U_0(\mathbf{x}, \mathbf{d})$.

The first step is to notice that all the paths share the same prefix and write it out explicitly:

$$U_\ell(\mathbf{x}_\ell, \mathbf{d}) = \bigoplus_{P \in \mathcal{P}_\ell(\mathbf{x}_\ell, \mathbf{d})} \bigodot_{s=\ell}^{L} m_s(A_s(\mathbf{x}_s^P), A'_s(\mathbf{x}_s^P - k_s^\ell(\mathbf{d})))$$

$$= \bigoplus_{P \in \mathcal{P}_\ell(\mathbf{x}_\ell, \mathbf{d})} m_\ell(A_\ell(\mathbf{x}_\ell), A'_\ell(\mathbf{x}_\ell - \mathbf{d})) \odot \left[ \bigodot_{s=\ell+1}^{L} m_s(A_s(\mathbf{x}_s^P), A'_s(\mathbf{x}_s^P - k_s^\ell(\mathbf{d}))) \right]. \qquad (6)$$

Now, we want to pull the prefix $m_\ell(A_\ell(\mathbf{x}_\ell), A'_\ell(\mathbf{x}_\ell - \mathbf{d}))$ out of the "sum". For that purpose, we will need the set of endpoints $\mathbf{out}(\mathbf{x}_\ell)$, introduced in the notation in Section 2. The "sum" can be re-written in terms of those endpoints as

$$U_\ell(\mathbf{x}_\ell, \mathbf{d}) = \bigoplus_{\substack{\mathbf{x}_{\ell+1} \in \mathbf{out}(\mathbf{x}_\ell) \\ P \in \mathcal{P}_{\ell+1}(\mathbf{x}_{\ell+1}, \gamma_{\ell+1}(\mathbf{d}))}} m_\ell(A_\ell(\mathbf{x}_\ell), A'_\ell(\mathbf{x}_\ell - \mathbf{d})) \odot \left[ \bigodot_{s=\ell+1}^{L} m_s(A_s(\mathbf{x}_s^P), A'_s(\mathbf{x}_s^P - k_s^\ell(\mathbf{d}))) \right]. \qquad (7)$$

---

**Algorithm 1** Backward pass

---

1: **procedure** BACKWARD($G$, $G'$)
2:     **for** $A_L(\mathbf{x}_L) \in A_L$ **do**
3:         **for** $\mathbf{d} \in k_L(\mathcal{D})$ **do**
4:             $U_L(\mathbf{x}_L, \mathbf{d}) \leftarrow m_L(A_L(\mathbf{x}_L), A'_L(\mathbf{x}_L - \mathbf{d}))$,                    ▷ Initialize the last layer.
5:         **end for**
6:     **end for**
7:     **for** $\ell$ = L-1, ..., 0 **do**
8:         **for** $A_\ell(\mathbf{x}_\ell) \in A_\ell$ **do**
9:             **for** $\mathbf{d} \in k_\ell(\mathcal{D})$ **do**
10:                 $S \leftarrow 0$,
11:                 **for** $\mathbf{x}_{\ell+1} \in \mathbf{out}(\mathbf{x}_\ell)$ **do**
12:                     $S \leftarrow S \oplus U_{\ell+1}(\mathbf{x}_{\ell+1}, \gamma_{\ell+1}(\mathbf{d}))$,
13:                 **end for**
14:                 $U_\ell(\mathbf{x}_\ell, \mathbf{d}) \leftarrow m_\ell(A_\ell(\mathbf{x}_\ell), A'_\ell(\mathbf{x}_\ell - \mathbf{d})) \odot S$,
15:             **end for**
16:         **end for**
17:     **end for**
18:     **return** $U_0$                            ▷ Return the distribution for the first layer.
19: **end procedure**

---

The last step is to use the left-distributivity of $\odot$ over $\oplus$ to pull the prefix out of the "sum":

$$U_\ell(\mathbf{x}_\ell, \mathbf{d}) = m_\ell(A_\ell(\mathbf{x}_\ell), A'_\ell(\mathbf{x}_\ell - \mathbf{d})) \odot \bigoplus_{\substack{\mathbf{x}_{\ell+1}\in\mathbf{out}(\mathbf{x}_\ell) \\ P\in\mathcal{P}_{\ell+1}(\mathbf{x}_{\ell+1}, \gamma_{\ell+1}(\mathbf{d}))}} \bigodot_{s=\ell+1}^{L} m_s(A_s(\mathbf{x}_s^P), A'_s(\mathbf{x}_s^P - k_s^\ell(\mathbf{d})))$$

$$= m_\ell(A_\ell(\mathbf{x}_\ell), A'_\ell(\mathbf{x}_\ell - \mathbf{d})) \odot \bigoplus_{\mathbf{x}_{\ell+1}\in\mathbf{out}(\mathbf{x}_\ell)} U_{\ell+1}(\mathbf{x}_{\ell+1}, \gamma_{\ell+1}(\mathbf{d})). \tag{8}$$

The detailed procedure is listed in Algorithm 1. We use the notation $k_\ell(\mathcal{D})$ for the set of subsampled shifts which is the result of applying function $k_\ell$ to every element of the set of initial shifts $\mathcal{D}$.

## 5   Choice of neuron matching function $m$ and operators $\oplus, \odot$

For the convolutional layers, we use the matching function

$$m_{\mathtt{conv}}(w, v) = \begin{cases} 0 & \text{if } w = 0, v = 0, \\ \frac{\min(w,v)}{\max(w,v)} & \text{otherwise.} \end{cases} \tag{9}$$

For the max-pooling layers, the computational graph can be truncated to just one active connection (as only one element influences higher-level features). Moreover, max-pooling does not create any additional features, only passes/subsamples the existing ones. Thus it does not make sense to take into account the pre-activations for those layers as they are the same as activations (up to subsampling). For these reasons, we use

$$m_{\mathtt{maxpool}}(w, v) = \delta(w = \arg\max N_w) \wedge \delta(v = \arg\max N_v), \tag{10}$$

where $N_w$ is the neighborhood of max-pooling covering node $w$, $\delta(\cdot)$ is the indicator function (1 if the condition holds, 0 otherwise).

In this paper, we use sum as $\oplus$ and product as $\odot$. Another possible choice would be $\max$ for $\oplus$ and $\min$ or product for $\odot$ — theoretically, those combinations satisfy the conditions in Theorem 1. Nevertheless, we found sum/product combination working better than others. This could be explained by the fact that $\max$ as $\oplus$ would be taken over a huge set of paths which is not robust in practice.

## 6   Experiments

We validate our approach in the field of computer vision as our method requires a convolutional neural network trained on a large recognition dataset. Out of the vision correspondence tasks, we

Table 1: Summary of the convolutional neural network VGG-16. We only show the part up to the 8-th layer as we do not use higher activations (they are not pixel-related enough). In the layer type row, $c$ stands for 3x3 convolution with stride 1 followed by the ReLU non-linear activation function [11] and $p$ for 2x2 max-pooling with stride 2. The input to convolution is padded with the "same as boundary" rule.

| Layer index | 1 | 2 | 3 | 4 | 5 | 6 | 7 | 8 |
|---|---|---|---|---|---|---|---|---|
| Layer type | c | c | p | c | c | p | c | c |
| Output channels | 64 | 64 | 64 | 128 | 128 | 128 | 256 | 256 |

chose stereo matching to validate our method. For this task, the input data dimensionality is $B = 2$ and the shift set is represented by horizontal shifts $\mathcal{D} = \{(0, 0, 0), \dots, (D_{\max}, 0, 0)\}$. We always convert images to grayscale before running CNNs, following the observation by [25] that color does not help.

For pre-trained recognition CNN, we chose the VGG-16 network [20]. This network is summarized in Table 1. We will further refer to layer indexes from this table. It is important to mention that we have not used the whole range of layers in our experiments. In particular, we usually started from layer 2 and finished at layer 8. As such, it is still necessary to consider multi-channel input. To extend our algorithm to this case, we create a virtual input layer with $C = 1$ and virtual per-pixel arcs to all the real input channels. While starting from a later layer is an empirical observation which improves the results for our method, the advantage of finishing at an earlier layer was discovered by other researchers as well [5] (starting from some layer, the network activations stop being related to individual pixels). We will thus abbreviate our methods as "ours(s, t)" where "s" is the starting layer and "t" is the last layer.

## 6.1 Experimental setup

For the stereo matching, we chose the largest available datasets KITTI 2012 and KITTI 2015. All image pairs in these datasets are rectified, so correspondences can be searched in the same row. For each training pair, the ground-truth shift is measured densely per-pixel. This ground truth was obtained by projecting the point cloud from LIDAR on the reference image. The quality measure is the percentage $\texttt{Err}_t$ of pixels whose predicted shift error is bigger than a threshold of $t$ pixels. We considered a range of thresholds $t = 1, \dots, 5$, while the main benchmark measure is $\texttt{Err}_3$. This measure is only computed for the pixels which are visible in both images from the stereo pair.

For comparison with the baselines, we used the setup proposed in [25] — the seminal work which introduced deep learning for stereo matching and which currently stays one of the best methods on the KITTI datasets. [24] is an extensive study which has a representative comparison of learning-based and non-learning-based methods under the same setup and open-source code [24] for this setup. The whole pipeline works as follows. First, we obtain the raw scores $U(\mathbf{x}, \mathbf{d})$ from Algorithm 1 for the shifts up to $D_{\max} = 228$. Then we normalize the scores $U(\mathbf{x}, \cdot)$ per-pixel by dividing them over the maximal score, thus turning them into the range $[0, 1]$, suitable for running the post-processing code [24]. Finally, we run the post-processing code with exactly the same parameters as the original method [25] and measure the quality on the same 40 validation images.

## 6.2 Baselines

We have two kinds of baselines in our evaluation: those coming from [25] and our simpler versions of deep feature transfer similar to [13], which do not consider paths.

The first group of baselines from [25] are the following: the sum of absolute differences "sad", the census transform "cens" [23], the normalized cross-correlation "ncc". We also included the learning-based methods "fst" and "acrt" [25] for completeness, although they use training data to learn features while our method does not.

For the second group of baselines, we stack up the activation volumes for the given layer range and up-sample the layer volumes if they have reduced resolution. Then we compute normalized cross-correlation of the stacked features. Those baselines are denoted "corr(s, t)" where "s" is the

Table 2: This table shows the percentages of erroneous pixels $\text{Err}_t$ for thresholds $t = 1, \ldots, 5$ on the KITTI 2012 validation set from [25]. Our method is denoted "ours(2, 8)". The two right-most columns "fst" and "acrt" correspond to learning-based methods from [25]. We give them for completeness, as all the other methods, including ours, do not use learning.

| Threshold | Methods | | | | | | | | |
| | sad | cens | ncc | corr(1, 2) | corr(2, 2) | corr(2, 8) | ours(2, 8) | fst | acrt |
|---|---|---|---|---|---|---|---|---|---|
| 1 | - | - | - | 20.6 | 20.4 | 20.7 | **17.4** | - | - |
| 2 | - | - | - | 10.5 | 10.4 | 8.14 | **6.40** | - | - |
| 3 | 8.16 | 4.90 | 8.93 | 7.58 | 7.52 | 5.23 | **3.94** | 3.02 | 2.61 |
| 4 | - | - | - | 6.19 | 6.13 | 4.02 | **2.99** | - | - |
| 5 | - | - | - | 5.40 | 5.36 | 3.42 | **2.49** | - | - |

Table 3: KITTI 2012 ablation study.

| Threshold | Methods | | | |
| | ours(2, 2) | ours(2, 3) | central(2, 8) | ours(2, 8) |
|---|---|---|---|---|
| 1 | 17.7 | 18.4 | **17.3** | 17.4 |
| 2 | 7.90 | 8.16 | 6.58 | **6.40** |
| 3 | 5.28 | 5.41 | 4.02 | **3.94** |
| 4 | 4.08 | 4.05 | 3.04 | **2.99** |
| 5 | 3.41 | 3.32 | 2.53 | **2.49** |

starting layer, "t" is the last layer. Note that we correlate the features before applying ReLU following what [25] does for the last layer. Thus we use the input to the ReLU inside the layers.

All the methods, including ours, undergo the same post-processing pipeline. This pipeline consists of semi-global matching [9], left-right consistency check, sub-pixel enhancement by fitting a quadratic curve, median and bilateral filtering. We refer the reader to [25] for the full description. While the first group of baselines was tuned by [25] and we take the results from that paper, we had to tune the post-processing hyper-parameters of the second group of baselines to obtain the best results.

### 6.3 KITTI 2012

The dataset consists of 194 training image pairs and 195 test image pairs. The reflective surfaces like windshields were excluded from the ground truth.

The results in Table 2 show that our method "ours(2, 8)" performs better compared to the baselines. At the same time, its performance is lower than learning-based methods from [25]. The main promise of our method is scalability: while we test it on a task where huge effort was invested into collecting the training data, there are other important tasks without such extensive datasets.

### 6.4 Ablation study on KITTI 2012

The goal of this section is to understand how important is the deep hierarchy of features versus one or few layers. We compared the following setups: "ours(2, 2)" uses only the second layer, "ours(2, 3)" uses only the range from layer 2 to layer 3, "central(2, 8)" considers the full range of layers but only with central arcs in the convolutions (connecting same pixel positions between activations) taken into account in the backward pass, "ours(2, 8)" is the full method. The result in Table 3 shows that it is profitable to use the full hierarchy both in terms of depth and coverage of the receptive field.

### 6.5 KITTI 2015

The stereo dataset consists of 200 training image pairs and 200 test image pairs. The main difference to KITTI 2012 is that the images are colored and the reflective surfaces are present in the evaluation.

Similar conclusions to KITTI 2012 can be drawn from experimental results: our method provides a reasonable transfer, being inferior only to learning-based methods — see Table 4. We show our depth map results in Fig. 2.

Table 4: This table shows the percentages of erroneous pixels $\mathtt{Err}_t$ for thresholds $t = 1, \ldots, 5$ on the KITTI 2015 validation set from [25]. Our method is denoted "ours(2, 8)". The two right-most columns "fst" and "acrt" correspond to learning-based methods from [25]. We give them for completeness, as all the other methods, including ours, do not use learning.

| Threshold | Methods | | | | | | | | |
| | sad | cens | ncc | corr(1, 2) | corr(2, 2) | corr(2, 8) | ours(2, 8) | fst | acrt |
| --- | --- | --- | --- | --- | --- | --- | --- | --- | --- |
| 1 | - | - | - | 26.6 | 26.5 | 29.6 | **26.2** | - | - |
| 2 | - | - | - | 10.9 | 10.8 | 11.2 | **9.27** | - | - |
| 3 | 9.44 | 5.03 | 8.89 | 6.68 | 6.63 | 6.16 | **4.78** | 3.99 | 3.25 |
| 4 | - | - | - | 5.05 | 5.03 | 4.42 | **3.36** | - | - |
| 5 | - | - | - | 4.22 | 4.20 | 3.60 | **2.72** | - | - |

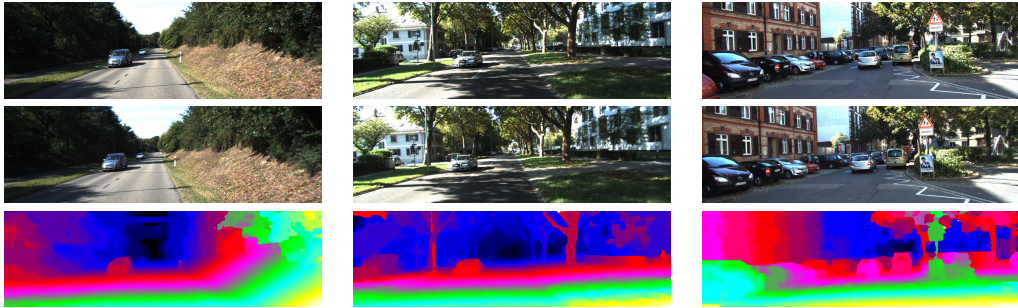

Figure 2: Results on KITTI 2015. Top to bottom: reference image, searched image, our depth result. The depth is visualized in the standard KITTI color coding (from close to far: yellow, green, purple, red, blue).

## 6.6  Style transfer experiment on KITTI 2015

The goal of this experiment is to show the robustness of recognition hierarchy for the transfer to correspondence search — something we advocated in the introduction as the advantage of our approach. We apply the style transfer method [5], implemented in the Prisma app. We ran different style transfers on the left and right images. While now very different at the pixel level, the higher level descriptions of the images remain the same which allows to successfully run our method. The qualitative results show the robustness of our path-based method in Fig. 3 (see also Fig. 2 for visual comparison to normal data).

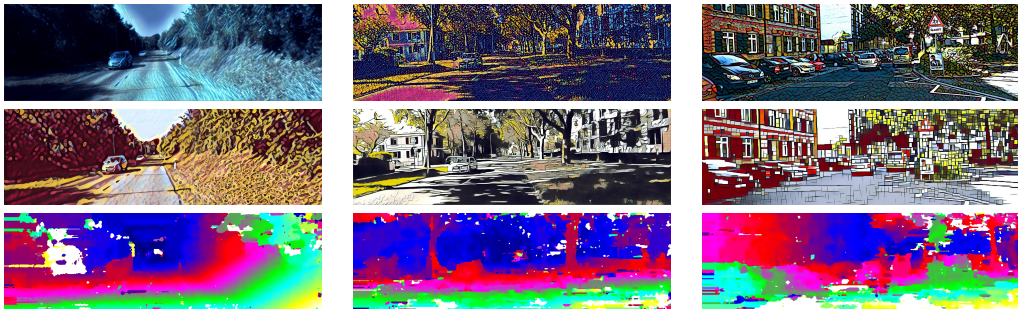

Figure 3: Results for the style transfer on KITTI 2015. Top to bottom: reference image, searched image, our depth result. The depth is visualized in the standard KITTI color coding (from close to far: yellow, green, purple, red, blue).

# 7 Conclusion

In this work, we have presented a method for transfer from recognition to correspondence search at the lowest level. For that, we re-use activation paths from deep convolutional neural networks and propose an efficient polynomial algorithm to aggregate an exponential number of such paths. The empirical results on the stereo matching task show that our method is competitive among methods which do not use labeled data from the target domain. It would be interesting to apply this technique to sound, which should become possible once a high-quality deep convolutional model becomes accessible to the public (e.g., [21]).

### Acknowledgements

We would like to thank Dmitry Laptev, Alina Kuznetsova and Andrea Cohen for their comments about the manuscript. We also thank Valery Vishnevskiy for running our code while our own cluster was down. This work is partially funded by the Swiss NSF project 163910 "Efficient Object-Centric Detection".

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
