[Reviews · NeurIPS 2017]

Reviewer 1



The work proposed a method for correspondence search at the lowest level (e.g., pixels in images) using the activation paths in neural networks (CNN in particular). The high level idea is that if two pixels are corresponding ones in a reference and a searched image if they lead to similar activation patterns (parallel paths) from deep neural networks. The work uses backward dynamic programming (polynomial time algorithm) to compute the matching score between two pixels with exponential number of such paths. Experimental results on the stereo matching task showcase that the proposed method is competitive among non-learning-based methods. The proposed method is intuitive. The experiments have been limited to stereo matching, finding camera correspondence. Will the proposed method work with finding correspondence of objects placed under different background? The authors suggested that using the full hierarchy of neural networks improves the performance of the proposed method. How big an impact does the structure of the neural network itself play in the performance? Whether or not it would be beneficial to re-tune the CNN on the images from the experiment tasks? Minor comment: 1. Please explain the figures in more details. What does the coloring in the third row in Figure 2 and 3 mean?

Reviewer 2



The paper proposes a method to use a pre-trained network for image parts matching. This is an interesting and new way to transfer knowledge learned for recognition to a the quite different task of correspondance search. The method aims at comparing the activation values of several layers for two images and searching for similar activation paths across the layers. Since the number of possible paths is exponential in the number of layers, this work introduces an algorithm to aggregate all of them and solve the correspondance search in polynomial time. + the paper is really well written and curated in the details + the method is new and sound + the experiments demonstrate the effectiveness of the proposed approach Among the methods that deal with correspondence and has a learning phase, this recent work C. B. Choy, J. Gwak, S. Savarese, M. Chandraker, Universal Correspondence Network, in Neural Information Processing Systems (NIPS), 2016. was not cited and should be discussed in the text. Since two other learning-based methods are considered in the experiments, also this one could be used as reference. One high level question: how is this approach dependend on the original pre-trained network? Here the paper builds over VGG16, but could it be used on other architectures (simples ones: alexnet, more complicated: Inception) and how would it influence the results? Among the

Reviewer 3



This paper presents a method for determining spatial correspondences between two images using a pre-trained convolutional network, without needing to train on the specific correspondence task. Given two images, the matching similarity at each pixel location is defined as the sum total similarity between all pairs of possible paths through network, starting at each location and its corresponding offset location in the other image. For a single offset value, this results in a recurrenceĀ relation dependent on the next layer up, which enables a dynamic-programming-like algorithm to compute all costs in linear time using a single backwards pass. In this way, a multi-layer matching is performed for each offset, accounting for a hierarchy of scales and features. This algorithm is applied to the task of stereo matching, where it achieves performance slightly lower than SOA supervised methods that require retraining, but better than a baseline computing correlation between stacked feature maps. This idea is an interesting way to use a pretrained network for additional uses. However, I think it could be further explored and evaluated, and perhaps more simply explained. In particular, I would like to see whether it can be applied to other dense matching tasks, e.g. video optical flow, and compared to other dense correspondence methods, such as Deep Flow and SIFT Flow (in both speed and accuracy). In addition, the notation seems a little overly complex and can be hard to follow (though I was able to understand it) --- I think it would be good to try simplifying this if possible. Overall, I think this is an interesting method, but feel it could be further explored for more image correspondence applications. Additional detail: Twice the paper says the derivation of the algorithm makes use of the "associative" property, but this is actually the distributive property (not associative).